# Ammonium Dinitramide as a Prospective N–NO_2_ Synthon: Electrochemical Synthesis of Nitro-*NNO*-Azoxy Compounds from Nitrosoarenes

**DOI:** 10.3390/molecules29235563

**Published:** 2024-11-25

**Authors:** Alexander S. Budnikov, Nikita E. Leonov, Michael S. Klenov, Mikhail I. Shevchenko, Tatiana Y. Dvinyaninova, Igor B. Krylov, Aleksandr M. Churakov, Ivan V. Fedyanin, Vladimir A. Tartakovsky, Alexander O. Terent’ev

**Affiliations:** 1N. D. Zelinsky Institute of Organic Chemistry, Russian Academy of Sciences, 47 Leninsky Prosp., 119991 Moscow, Russia; alsbudnikov@gmail.com (A.S.B.); leonovne@ioc.ac.ru (N.E.L.); mishashev4enko@yandex.ru (M.I.S.); tatianadvinyaninova@yandex.ru (T.Y.D.); churakov@ioc.ac.ru (A.M.C.); tva@ioc.ac.ru (V.A.T.); terentev@ioc.ac.ru (A.O.T.); 2Higher Chemical College of the Russian Academy of Sciences, D. I. Mendeleev University of Chemical Technology of Russia, 9 Miusskaya Square, 125047 Moscow, Russia; 3A. N. Nesmeyanov Institute of Organoelement Compounds, Russian Academy of Sciences, 28 Vavilova Str., 119991 Moscow, Russia; octy@xrlab.ineos.ac.ru

**Keywords:** electrochemistry, N–N coupling, azoxy compounds, ammonium dinitramide, nitroso compounds, antifungal agents

## Abstract

In this study, the electrochemical coupling of nitrosoarenes with ammonium dinitramide is discovered, leading to the facile construction of the nitro-*NNO*-azoxy group, which represents an important motif in the design of energetic materials. Compared to known approaches to nitro-*NNO*-azoxy compounds involving two chemical steps (formation of azoxy group containing a leaving group and its nitration) and demanding expensive, corrosive, and hygroscopic nitronium salts, the presented electrochemical method consists of a single step and is based solely on nitrosoarenes and ammonium dinitramide. The dinitramide salt plays the roles of both the electrolyte and reactant for the coupling. Despite the fact that many side reactions can be expected due to the redox-activity of both the reagents and target products, under optimized conditions the synthesis is performed in an undivided cell under constant current conditions with high current density and can be easily scaled up without a reduction in the product yield. Moreover, the synthesized nitro-*NNO*-azoxy compounds are discovered to be potent fungicides active against a broad range of phytopathogenic fungi.

## 1. Introduction

Organic compounds with N–N and N–O bonds demonstrate a huge diversity of structures, redox properties [1,2], and practical applications. Relatively weak N–O bonds contribute to the diverse applications of such compounds as precursors of free radicals for selective synthetic transformations [2,3,4,5,6] or polymerization initiation [7]. Organic molecules containing nitrogen–oxygen systems are useful energetic materials [8], HNO- and NO-donors [9,10], antiaggregant agents [11,12], and fungicides [13]. On the other hand, there is a clear lack of effective synthetic approaches towards such compounds owing to the numerous side processes possible for nitrogen–oxygen systems due to their rich reactivity. In particular, a very limited number of N–N [2,14,15,16,17,18] or N–O coupling methods [19,20,21] exist, in contrast to the numerous methods for C–C and C-Heteroatom coupling. These processes are challenging and highly valued not only due to their possible side processes, which have to be suppressed by careful and insightful reaction design, but also due to the quite low thermodynamic driving force for the formation of the N–N and N–O bonds. In the basis of the present work, we put forward the idea of developing an organic electrosynthesis methodology to use electrical energy for the realization of such thermodynamically-challenging syntheses in a green and versatile way.

Nowadays, electro-organic synthesis has become one of the leading fields in modern organic chemistry [22,23,24]. A sole and cheap electric current as a “traceless” reagent provides an attractive alternative to expensive and sometimes toxic chemical oxidants and reductants [25]. The primary challenge of electro-organic synthesis lies in achieving selectivity, as the potential for the overoxidation or reduction of the starting materials restricts the range of substrates that can be effectively utilized. The electrochemical formation of N–N and N=N bonds remains largely poorly examined within the field, presenting a significant opportunity for research and development [26,27]. While various electrochemical processes have focused mainly on C–N bond formation [28,29,30,31,32], the specific mechanisms and methodologies for N–N bond synthesis have not been extensively explored. In recent years, several elegant approaches for electrochemical N–N/N=N bond formation have been developed [33,34,35,36,37,38,39,40,41,42], although the majority of these studies are limited to intramolecular radical cyclizations [26,33,34,36,37,39,40] rather than intermolecular [35,38,41,42] coupling. It is apparent that intermolecular reactions involving reactive intermediates are difficult to realize compared to intramolecular reactions, but on the other hand intermolecular transformations are usually based on more synthetically available starting materials and open access to a wider structural diversity of target products. Therefore, the development of electrochemical intermolecular N–N/N=N bond-forming methods is desirable.

An additional approach to the problem of the construction of nitrogen–oxygen systems is the usage of synthetically available building blocks already containing N–N and N–O bonds. In the present work, we propose ammonium dinitramide (NH_4_N(NO_2_)_2_), a compound available on an industrial scale [43], as an efficient reagent for the electrochemical construction of the nitro-*NNO*-azoxy group via intermolecular N=N bond formation on the basis of nitrosoarenes (Figure 1). Nitro-*NNO*-azoxy compounds were first designed and synthesized at ZIOC RAS by Churakov et al. [44,45,46]. In recent years, the nitro-*NNO*-azoxy group has attracted considerable attention as a useful fragment in the design of potential energetic materials with outstanding performance [47,48,49,50,51,52,53,54,55,56], some of which are among the most energetic compounds, such as TKX-50 [57], nitroazole fused 1,2,3,4-tetrazines [58,59], *N*-azo nitro-1,2,3-triazoles [60], dinitropyrazolo-triazine (PTX) [61], bi-1,2,4-triazole with trinitromethyl groups [62], and azido and tetrazolo 1,2,4,5-tetrazine *N*-oxides [63]. To date, there is only one two-step synthetic approach to the introduction of a nitro-*NNO*-azoxy group into the benzene ring (Figure 1). The first step is the coupling between ArNO and Br_2_NX (X = Ac [44], *^t^*Bu [45], CO_2_*^t^*Bu [45], and C(O)NH_2_ [45]), leading to azoxy compounds containing a leaving group X. The second step includes the substitutive nitration of the latter with strong nitrating reagents (NO_2_BF_4_, (NO_2_)_2_SiF_6_, and N_2_O_5_). The disadvantages of this approach are the use of expensive and hazardous nitronium salts and its inefficiency for the synthesis of (nitro-*NNO*-azoxy)arenes with electron-donating substituents due to the numerous side reactions at the nitration stage. Therefore, the development of a novel synthetic route to these compounds, devoid of the shortcomings of the known approach, represents a prime challenging task.

Based on data on the potential biocidal activity of azoxy compounds [64,65,66,67], synthesized nitro-*NNO*-azoxy arenes were tested for in vitro activity against phytopathogenic fungi, which pose a serious threat to global crop production. The observed pronounced fungicidal activity demonstrates that nitro-*NNO*-azoxy arenes are a prospective new structural class for fungicide development. This discovery is particularly crucial given the ongoing emergence of phytopathogenic fungal strains with resistance to conventional synthetic fungicides [68,69,70].

## 2. Results and Discussion

Electrochemical introduction of nitro-*NNO*-azoxy fragment was discovered by employing nitrosobenzene **1a** and ammonium dinitramide (ADN) as starting reagents. An extensive optimization of the reaction parameters (including supporting electrolyte, solvent system, additives, the amount of electricity passed, current, and electrode materials) was conducted to evaluate the feasibility of the discovered transformation (Table 1).

The comprehensive optimization of the reaction conditions (see Appendix A in Appendix A for more details) revealed that performing the reaction under CCE conditions using a carbon felt anode and platinum wire cathode with 4 equiv. of ADN as the reagent and supporting electrolyte delivered the desired nitro-*NNO*-azoxybenzene **2a** in 85% yield (70% of isolated product, Table 1, entry 1). Employing 1 equivalent of ADN in the presence of *n*-Bu_4_NBF_4_ as a supporting electrolyte afforded **2a** only in a 11% yield (Table 1, entry 2). Along with the formation of the target product **2a**, the formation of a complex mixture of nitrosobenzene oxidation or reduction products, such as nitrobenzene and azoxybenzene, was observed. We envisioned that ADN itself could act as a supporting electrolyte with ammonium reduction at the cathode surface, therefore suppressing the competitive reduction [71] of **1a**. The evaluation of dinitramide loading revealed 4 mmol per mmol **1a** as an optimal amount (Table 1, entry 3). The replacement of ADN by its potassium salt (KDN) in the presence of supporting electrolyte (necessary due to insufficient KDN solubility) delivered **2a** in a 34% yield (Table 1, entry 4). Entry 5 proved that acetonitrile was the optimal solvent for the discovered transformation. The optimal amount of electricity for the reaction was found to be 2 F per mole of **1a** (Table 1, entry 6). Increasing the current density with a platinum anode scored a comparable yield; however, further increases reduced the yield of **2a** (Table 1, entry 7). We were delighted to find that by employing a carbon felt anode we were able to further increase the current density without significant loss of the yield (Table 1, entries 1, 8, 9). While these reaction conditions did not lead to a significant decrease of the yield of **2a**, sufficient heating of the reaction mixture was observed. Entries 10–11 showed that the best results were obtained with a carbon felt anode and platinum wire cathode. Finally, no reaction product was observed without an electric current (Table 1, entry 12).

With optimal conditions in hand, the scope of the developed electrochemical nitro-*NNO*-azoxylation protocol was evaluated (Figure 2).

The discovered electrochemical nitro-*NNO*-azoxylation was applicable to various substituted nitrosobenzenes **1a**–**g,i**–**t**, possessing both electron-withdrawing (**2e**–**g**) and electron-donating groups (**2b**–**d,i**–**k**), leading to the desired nitro-*NNO*-azoxybenzenes **2a**–**g,i**–**t** in good yields. Compounds **2a,e**–**g,o**–**q** were previously reported [44,45]. The substitution pattern in the aryl ring does not have a significant effect on the yield of **2** for both the acceptor and donor derivatives, which shows the versatility of the developed protocol. However, no reaction product **2h** was observed upon the introduction of a strongly electron-donating *N*,*N*-dimethylamino group; *N,N*-dimethyl-4-nitroaniline was isolated as the only reaction product. The halogen substituents on the aryl ring were also tolerated (products **2l**–**2t**). Higher yields were obtained for chlorine-substituted nitrosobenzenes (products **2m**–**p**, 52–80%) compared to bromine-substituted ones (products **2q**–**t**, 34–53%). In both cases, the best results were obtained for 2,4,6-trisubstituted derivatives (products **2p** and **2t**, 80 and 53%, respectively). It is worth noting that for a series of 4-substitutied derivatives, mainly **2b**, **2i**, **2l**, **2m** and **2q**, the formation of the corresponding substituted nitrobenzenes **2b′**, **2i′**, **2l′**, **2m′** and **2q′** was observed in noticeable quantities.

The synthetic utility of the discovered nitro-*NNO*-azoxylation protocol was shown by the 10 mmol scale synthesis of **2a** (Figure 3). The desired nitro-*NNO*-azoxy benzene **2a** was obtained in 65% yield (1.086 g, 6.5 mmol).

To clarify the reaction mechanism, cyclic voltammetry (CV) studies of **1a**, **1e**, **1i**, and ADN were conducted (Figure 1). Previously, the electrochemical behaviour of ADN was studied in dimethylsulfoxide [72] and water [73] solutions, as well as in a molten state [74].

The CV data indicate the expected correlation of oxidation potentials (given vs. Ag/AgNO_3_ in MeCN) for nitrosobenzenes with the electronic effect of substituent on the benzene ring: the lowest oxidation potential (+1.28 V, blue curve) is observed for **1i** with an electron-donating *p*-OMe group, while the highest oxidation potential (+1.90 V, pink curve) is observed for **1e** with an electron-withdrawing *p*-NO_2_ group. Unsubstituted nitrosobenzene **1a** is oxidized in between at +1.69 V (green curve). ADN exhibits two irreversible anodic peaks [72] (red curve) at +1.78 V and +2.04 V, which are higher in comparison with **1i** and **1a** and comparable with **1e**.

Upon the addition of ADN to the PhNO solution (Figure 2, yellow curve), the peak of **1a** oxidation shifts to the lower potential region (+1.46 V), while peaks corresponding to the ADN oxidation are slightly shifted to higher potentials (+1.92 V and +2.08 V). Therefore, the direct anodic oxidation of nitrosobenzene **1a** as an initial step of the discovered transformation cannot be ruled out. Finally, the reaction product nitro-*NNO*-azoxy benzene **2a** is electrochemically inert in the range of 0–2.3 V (Figure 2, blue curve); however, in a negative potential region, **2a** undergoes cathodic reduction (for more details, see Appendix A in Appendix A).

Next, the reaction mechanism was further investigated by conducting a series of control experiments (Figure 4).

Performing electrolysis in a divided H-cell with **1a** and ADN in an anodic chamber (Figure 4A) gave the corresponding nitro-*NNO*-azoxylation product **2a** in a 62% yield. This result indicates that the studied process occurs at an anodic surface. Monitoring the reaction potential during electrolysis in standard reaction conditions with a reference electrode (Figure 4B) revealed that the observed potential is in good correlation with the oxidation potential of ADN in its mixture with **1a** derived from the CV study (Figure 2, yellow curve). According to the CV data (see Figure 1 and Figure 2), nitrosobenzene **1a** is more easily oxidized (1.69 V alone and 1.46 V in the reaction mixture) than ADN (1.78 V alone and 1.92 V in the reaction mixture), suggesting the anodic oxidation of **1a** as a plausible first step towards **2a**. Thus, the reaction was carried out under constant potential electrolysis (CPE) conditions at E∼1.8 V for 2.0 F per mole of **1a** to verify this mechanistic proposal (Figure 4C). Under CPE conditions, the formation of **2a** was observed in a low yield of 15%. However, when we applied an excess amount of electricity (5.5 F per mole of **1a**), **2a** was formed in 52% yield. Thus, we speculate that the oxidation of nitrosobenzene **1a** occurring at a potential of 1.69 V (or 1.46 V) practically does not lead to the target transformation, and the oxidation of ADN is crucial for the studied process. It should be noted that ADN excess expectedly favours its anodic oxidation (see Appendix A). To gain insight into ADN transformation in the discovered nitro-*NNO*-azoxylation, the electrolysis of ADN in the absence of **1a** was conducted in undivided and divided cells under standard reaction conditions (Figure 4D). The ammonium nitrate was observed in both cases as the only detectable product. Moreover, in an undivided cell, the ratio of the recovered ADN and ammonium nitrate was 1:2 and the total mass decreased by more than half. Electrolysis in a divided electrochemical cell revealed that in the cathodic chamber, the recovery of ADN was 85% and no other products of cathodic reduction of ADN were observed [72,73,74]. At the same time, after the completion of electrolysis and evaporation of the MeCN, the anodic chamber contained only ammonium nitrate (27.5% from ADN loading). These results indicate that ADN is capable of deeper oxidation at the anodic surface with the formation of gaseous products.

Based on the obtained results from the conducted CV studies, control experiments, and literature data about the electrochemical behaviour of nitrosobenzene and ADN, the following mechanism of electrochemical synthesis of nitro-*NNO*-azoxy compounds **2** from nitrosoarenes **1** was proposed (Figure 5).

First, direct anodic oxidation of the dinitramide anion provides *N*-centred radical **A**, which reacts with nitrosoarene **1** with the formation of *N*-oxyl radical **C**. However, we do not exclude a competitive path with oxidation of **1** to cation-radical **B** and its subsequent reaction with dinitramide anion leading to the same radical **C**. This pathway could be realized for nitrosobenzenes containing electron-donating groups in the benzene ring, such as OMe (see Figure 1). Radical **C** undergoes NO_2_ extrusion, with the formation of nitro-*NNO*-azoxy compound **2**. An alternative pathway for the formation of **2** is NO_2_ extrusion from **A**, leading to nitro-nitrene **D**, which reacts with nitroso compound **1**, giving **2**.

## 3. In Vitro Fungicidal Activity of the Synthesized Nitro-*NNO*-Azoxy Compounds

Nowadays, phytopathogenic fungi are one of the most dangerous threats to crop production and public health [75,76,77,78,79]. In this regard, fungicides remain the most effective means for protecting agricultural crops from fungal diseases caused by produced mycotoxins [80]. However, most fungicides available on the global market fall into a limited number of classes and share common mechanisms of action, leading to the development of fungicidal resistance in both agriculture and medicine against these known active compound classes [68,69,70]. Therefore, the discovery of new fungicidal classes with unforeseen mechanism of action is a primary scientific goal.

In the second part of our study, synthesized nitro-*NNO*-azoxy compounds **2** were tested for fungicidal activity against 6 phytopathogenic fungi from different taxonomic classes that pose a high threat to agricultural crop yields: Venturia inaequalis (*V.i*.), Sclerotinia sclerotiorum (*S.s.*), Fusarium oxysporum (*F.o*.), Fusarium moniliforme (*F.m.*), Bipolaris sorokiniana (*B.s.*), and Rhizoctonia solani (*R.s*.) (Table 2). The commercially available fungicide Triadimefon was used as the reference compound.

As can be seen from Table 2, compounds **2a**, **2b**, **2m**, **2o**, and **2s** exhibit the greatest activity against phytopathogenic fungi, superior to that of triadimefon. Compounds with electron-donating methyl and methoxy groups (**2b**–**d**, **2i**–**k**) also demonstrate pronounced fungicidal activity, whereas nitro-*NNO*-azoxy compounds bearing electron-withdrawing nitro groups (**2e**–**g**) are less active. Between bromine- and chlorine-containing derivatives (compounds **2m**–**t**), *ortho*-dihalogenated products (**2o** and **2s**) and *para*-chloro substituted product **2m** show superior fungicidal activity.

## 4. Materials and Methods

We caution that dinitramide salts should be treated carefully as shock-sensitive explosive substances, the stability of which can depend on the preparation method and sample quality; mixtures with reducing agents can be explosive. Nitrosoarenes must be handled with care as potentially toxic and carcinogenic compounds.

In all experiments, RT means 22–25 °C. The ^1^H, ^13^C, ^14^N, and ^15^N NMR spectra were recorded with Bruker spectrometers DRX-500 (500.1, 125.8, 36.1, 50.7 MHz, respectively) and AV600 (600.1, 150.9, 43.4, 60.8 MHz, respectively). Chemical shifts are reported in delta (*δ*) ppm units downfield from TMS (^1^H and ^13^C) or CH_3_NO_2_ (^14^N and ^15^N, negative values of δ_N_ correspond to upfield shifts). The IR spectra were recorded with a Bruker ALPHA-T spectrometer in the range 400–4000 cm^−1^ (resolution 2 cm^−1^) as a thin layer or as KBr pellets. High-resolution ESI mass spectra (HRMS) were recorded with a Bruker micrOTOF II instrument. Silica gel 60 Merck (15–40 μm) was used for the preparative column and thin-layer chromatography. Silica gel “Silpearl UV 254” was used for the preparative column and thin-layer chromatography. Analytical thin-layer chromatography (TLC) was carried out on Merck silica gel 60 F254 and “Silufol” TLC silica gel UV-254 aluminium sheets. All reagents were purchased from Acros and Sigma-Aldrich. Solvents were purified before use, according to standard procedures. MeCN was distilled over P_2_O_5_ and then over CaH_2_, acetone was distilled over KMnO_4_, MeOH was distilled over magnesium and iodine, and DMF was distilled over CaH_2_. 1,1,1,3,3,3-Hexafluoro-2-propanol (HFIP) was used as is from commercial sources. All other reagents were used without further purification. Ammonium dinitramide [81], nitrosobenzene (**1a**) [82], 1-methyl-4-nitrosobenzene (**1b**) [83], 1-methyl-3-nitrosobenzene (**1c**) [84], 1-methyl-2-nitrosobenzene (**1d**) [85], 1-nitro-4-nitrosobenzene (**1e**) [86], 1-nitro-3-nitrosobenzene (**1f**) [87], 1-nitro-2-nitrosobenzene (**1g**) [88], 1-methoxy-4-nitrosobenzene (**1i**) [89], 1-methoxy-3-nitrosobenzene (**1j**) [90], 1-methoxy-2-nitrosobenzene (**1k**) [91], 1-fluoro-4-nitrosobenzene (**1l**) [92], 1-chloro-4-nitrosobenzene (**1m**) [84], 2,4-dichloro-1-nitrosobenzene (**1n**) [93], 1,3-dichloro-2-nitrosobenzene (**1o**) [94], 1,3,5-trichloro-2-nitrosobenzene (**1p**) [94], 1-bromo-4-nitrosobenzene (**1q**) [95], 2,4-dibromo-1-nitrosobenzene (**1r**) [96], 1,3-dibromo-1-nitrosobenzene (**1s**) [94], and 1,3,5-tribromo-1-nitrosobenzene (**1t**) [94] were prepared according to the reported procedures. All electrodes except for the carbon felt (C_F_) electrodes were polished flat plates with equal real surface areas and geometrical surface areas. Commercial carbon felt (PANCF3200300, produced from polyacrylonitrile, carbon content ≥98%, 3 mm thickness) was used as is. Electrode surface area is one of the key factors affecting the selectivity of electrode processes [97] due to the dependence of electrochemical potential on current density and other factors. The carbon felt electrode surface was estimated by the integration of cyclic voltammograms (15 cycles were done to achieve constant voltammogram) of ferrocene (0.001 M) in MeCN containing *n*-Bu_4_NBF_4_ (0.1 M) as the supporting electrolyte for each carbon felt sample and glassy carbon plates with known surface area were used as working electrodes. The specific surface area of the carbon felt samples estimated this way was about 0.05 m^2^·g^−1^ (carbon felt density was 0.14 g·cm^−3^).


**General Procedure for the Optimization of the Reaction Conditions for the Synthesis of (Nitro-*NNO*-azoxy)benzene (2a) from Nitrosobenzene (1a) (Experimental details for Table 1)**


An undivided 20 mL jacketed electrochemical cell was equipped with a platinum, stainless steel, nickel, graphite, carbon felt, or glassy carbon plate anode (30 × 15 mm^2^) and a platinum, stainless steel, nickel, graphite, glassy carbon plate (30 × 15 mm^2^), or platinum wire anode, and connected to a DC regulated power supply. Electrodes were completely immersed in the solution, giving S = 4.5 cm^2^ of working surface. A solution of nitrosobenzene (**1a**) (1.0 mmol, 107 mg) and ammonium dinitramide (ADN) (1–5 equiv., 1–5 mmol, 124–620 mg) in 20 mL of MeCN, acetone, DMF, MeOH, or HFIP was electrolyzed using constant current conditions (*I* = 15–240 mA) at 23–25 °C under magnetic stirring. After passing 1–3 F∙mol^−1^ of electricity (reaction time 14–322 min), electrodes were washed with CH_2_Cl_2_ (3 × 20 mL). The combined organic phase was washed with H_2_O (20 mL) and brine (20 mL), dried over Na_2_SO_4_, and the solvent removed in vacuo. The yields of **2a** were determined by ^1^H NMR spectroscopy using 2-methyl-2-nitropropane as an internal standard and checked by ^14^N NMR. In run 1, (nitro-*NNO*-azoxy)benzene (**2a**) was purified by preparative column chromatography on silica gel (*R*_f_ = 0.51, petroleum ether/ethyl acetate, 10:1) to give compound **2a** (142 mg, 85%) as a colourless solid. mp: 83–85 °C. The obtained product was identical (TLC, ^1^H, ^13^C, and ^14^N NMR) to the compound prepared according to the reported procedure [44].

**1-(nitro-*NNO*-azoxy)benzene (2a)**: White crystals, m.p. 26–27 °C (lit. m.p. 26–27 °C) [44], 75% yield. *R*_f_ (petroleum ether/ethyl acetate, 10:1) = 0.51. ^1^H NMR (500.13 MHz, CDCl_3_)*δ*: 7.57 (t, 2H, H(3), H(5), ^3^*J*_HH_ = 8.2 Hz), 7.21 (t, 1H, H(4), ^3^*J*_HH_ = 7.5 Hz), 8.10 (dd, 2H, H(2), H(6), ^3^*J*_HH_ = 8.8 Hz, ^4^*J*_HH_ = 1.1 Hz) ppm. ^13^C NMR (125.76 MHz, CDCl_3_)*δ*: 122.2 (s, C(2), C(6)), 129.8 (s, C(3), C(5)), 134.7 (s, C(4)), 142.8 (br. s, C(1)) ppm. The ^1^H–^13^C HSQC and HMBC experiments were used to assign the signals. ^14^N NMR (36.14 MHz, CDCl_3_)*δ*: −32 (N(O)=N–NO_2_, ∆ν_½_ = 20 Hz), −43 (N(O)=N–NO_2_, ∆ν_½_ = 130 Hz) ppm. IR (KBr): ν = 2955 (w), 2884 (w), 1635 (s), 1505 (w), 1328 (w), and 1282 (s) cm^−1^. Elemental analysis calcd (%) for C_6_H_5_N_3_O_3_: C 43.12, H 3.02, and N 25.14; found: C 43.15, H 3.03, and N 25.01.


**Typical Procedure for Electrochemical Synthesis of Nitro-*NNO*-Azoxy Benzenes 2a–2t (Experimental details for Figure 2)**


An undivided 20 mL jacketed electrochemical cell was equipped with a carbon felt anode (30 × 15 mm^2^) and a platinum wire cathode and connected to a DC regulated power supply. Electrodes were completely immersed in the solution, giving S = 4.5 cm^2^ of working anode surface. A mixture of nitrosobenzene **1a**–**1t** (1.0 mmol, 107–329 mg), and ADN (4.0 equiv., 4.0 mmol, 496 mg) in MeCN (20 mL) was electrolyzed using constant current conditions (*I* = 60 mA) at 23–25 °C under magnetic stirring. After passing 2.0 F of electricity per mole of **1a–1t** (reaction time 54 min), electrodes were washed with CH_2_Cl_2_ (3 × 20 mL). The combined organic phase was washed with H_2_O (20 mL) and brine (20 mL), dried over Na_2_SO_4_, and solvent was removed in vacuo. Products **2a**–**2t** were isolated by column chromatography on silica gel.

**1-Methyl-4-(nitro-*NNO*-azoxy)benzene (2b)**: Orange oil, 80% yield. *R*_f_ (petroleum ether/ethyl acetate, 40:1) = 0.25. ^1^H NMR (500.13 MHz, CDCl_3_)*δ*: 2.48 (s, 3H, Me), 7.35 (d, 2H, H(2), H(6), ^3^*J*_HH_ = 8.2 Hz), 7.99 (d, 2H, H(3), H(5), ^3^*J*_HH_ = 8.2 Hz) ppm. ^13^C NMR (125.76 MHz, CDCl_3_)*δ*: 21.1 (s, Me), 121.5 (s, C(3), C(5)), 129.7 (s, C(2), C(6)), 139.9 (br. s, C(4)), 145.6 (s, C(1)) ppm. The ^1^H–^13^C HSQC and HMBC experiments were used to assign the signals. ^14^N NMR (36.14 MHz, CDCl_3_)*δ*: −33 (N(O)=N–NO_2_, ∆ν_½_ = 20 Hz), −45 (N(O)=N–NO_2_, ∆ν_½_ = 130 Hz) ppm. IR (KBr): ν = 2952 (w), 2889 (w), 1611 (s), 1519 (w), 1478 (s), 1383 (w), 1345 (w), 1318 (w), 1268 (s), 1181 (m), and 1116 (w) cm^−1^. Elemental analysis calcd (%) for C_7_H_7_N_3_O_3_: C 46.41, H 3.90, and N 23.20; found: C 46.45, H 3.93, and N 23.01.

**1-Methyl-3-(nitro-*NNO*-azoxy)benzene (2c)**: Orange oil, 74% yield. *R*_f_ (petroleum ether/ethyl acetate, 40:1) = 0.50. ^1^H NMR (500.13 MHz, CDCl_3_)*δ*: 2.47 (s, 3H, Me), 7.45 (t, 1H, H(5), ^3^*J*_HH_ = 7.8 Hz), 7.52 (m, 1H, H(6)), 7.90–7.92 (m, 2H, H(2), H(4)) ppm. ^13^C NMR (125.76 MHz, CDCl_3_)*δ*: 21.3 (s, Me), 119.3 (s, C(4)), 122.5 (s, C(2)), 129.4 (s, C(5)), 135.3 (s, C(6)), 140.3 (s, C(1)), 142.7 (br. s, C(3)) ppm. The ^1^H–^13^C HSQC and HMBC experiments were used to assign the signals. ^14^N NMR (43.4 MHz, CDCl_3_)*δ*: −35 (N(O)=N–NO_2_, ∆ν_½_ = 15 Hz), −46 (N(O)=N–NO_2_, ∆ν_½_ = 100 Hz) ppm. IR (KBr): ν = 2924 (w), 2871 (w), 1610 (s), 1502 (s), 1470 (m), 1427 (w), 1383 (w), 1316 (w), 1295 (w), 1270 (s), 1220 (w), and 1169 (w) cm^−1^. Elemental analysis calcd (%) for C_7_H_7_N_3_O_3_: C 46.41, H 3.90, and N 23.20; found: C 46.46, H 3.91, and N 23.05.

**1-Methyl-2-(nitro-*NNO*-azoxy)benzene (2d)**: Orange oil, 71% yield. *R*_f_ (petroleum ether/ethyl acetate, 10:1) = 0.83. ^1^H NMR (600.13 MHz, CDCl_3_)*δ*: 2.51 (s, 3H, Me), 7.36–7.39 (m, 2H, H(4), H(6)), 7.52 (t, 1H, H(5), ^3^*J*_HH_ = 7.5 Hz), 7.69 (d, 1H, H(3), ^3^*J*_HH_ = 8.0 Hz) ppm. ^13^C NMR (150.90 MHz, CDCl_3_)*δ*: 19.1 (s, Me), 124.7 (s, C(3)), 127.9 (s, C(4)), 133.3 (s, C(1), C(6)), 133.4 (s, C(5)), 143.8 (br. s, C(2)) ppm. The ^1^H–^13^C HSQC and HMBC experiments were used to assign the signals. ^14^N NMR (43.4 MHz, CDCl_3_)*δ*: −34 (N(O)=N–NO_2_, ∆ν_½_ = 10 Hz), −41 (N(O)=N–NO_2_, ∆ν_½_ = 90 Hz) ppm. IR (KBr): ν = 2933 (w), 2878 (w), 1612 (s), 1524 (w), 1492 (m), 1473 (m), 1430 (w), 1385 (w), 1348 (w), 1302 (w), 1427 (w), 1273 (s), 1207 (w), 1163 (w), and 1148 (w) cm^−1^. Elemental analysis calcd (%) for C_7_H_7_N_3_O_3_: C 46.41, H 3.90, and N 23.20; found: C 46.50, H 3.94, and N 23.09.

**4-Nitro-1-(nitro-*NNO*-azoxy)benzene (2e)**: Pale yellow crystals, m.p. 49–50 °C (lit. m.p. 49–51 °C) [44], 79% yield. *R*_f_ (petroleum ether/ethyl acetate, 5:1) = 0.76. ^1^H NMR (500.13 MHz, CDCl_3_)*δ*: 8.39 (d, 2H, H(2), H(6), ^3^*J*_HH_ = 8.9 Hz), 8.49 (d, 2H, H(3), H(5), ^3^*J*_HH_ = 8.9 Hz) ppm. ^13^C NMR (150.9 MHz, CDCl_3_)*δ*: 123.9 (s, C(2), C(6)), 125.2 (s, C(3), C(5)), 146.4 (br. s, C(1)), 151.1 (br. s, C(4)) ppm. The ^1^H–^13^C HSQC and HMBC experiments were used to assign the signals. ^14^N NMR (36.14 MHz, CDCl_3_)*δ*: −17 (NO_2_, ∆ν_½_ = 155 Hz), −37 (N(O)=N–NO_2_, Δν_1/2_ = 20 Hz), −50 (N(O)=N–NO_2_, ∆ν_½_ = 120 Hz) ppm. IR (KBr): ν = 2873 (w), 1656 (m), 1631 (s), 1602 (s), 1539 (s), 1484 (m), 1407 (w), 1383 (w), 1319 (s), 1274 (m), 1172 (w), 1149 (w), 1125 (w), and 1108 (w) cm^−1^. Elemental analysis calcd (%) for C_6_H_4_N_4_O_5_: C 33.97, H 1.90, and N 26.41; found: C 33.99, H 1.94, and N 26.09.

**3-Nitro-1-(nitro-*NNO*-azoxy)benzene (2f)**: Pale yellow crystals, m.p. 53–54 °C (lit. m.p. 53–54 °C) [44], 46% yield. *R*_f_ (petroleum ether/ethyl acetate, 10:1) = 0.61.^1^H NMR (600.13 MHz, CDCl_3_)*δ*: 7.92 (t, 1H, H(5), ^3^*J*_HH_ = 8.3 Hz), 8.56 (d, 1H, H(4) or H(6), ^3^*J*_HH_ = 8.3 Hz), 8.64 (d, 1H, H(6) or H(4), ^3^*J*_HH_ = 8.3 Hz), 9.02 (s, 1H, H(2) ppm. ^13^C NMR (150.90 MHz, CDCl_3_)*δ*: 118.0 (s, C(2)), 127.8 (s, C(4) or C(6)), 129.0 (s, C(4) or C(6)), 131.3 (s, C(5)), 143.3 (br. s, C(1)), 148.5 (br. s, C(3)) ppm. The ^1^H–^13^C HSQC and HMBC experiments were used to assign the signals. ^14^N NMR (43.37 MHz, CDCl_3_)*δ*: −18 (NO_2_, ∆ν_½_ = 150 Hz), −38 (N(O)=N–NO_2_, ∆ν_½_ = 30 Hz), −51 (*N*(O)=N–NO_2_, ∆ν_½_ = 130 Hz) ppm. IR (KBr): ν = 2923 (w), 2877 (w), 2854 (w), 1625 (s), 1535 (s), 1493 (s), 1485 (m), 1434 (w), 1351 (s), 1322 (m), 1286 (m), 1265 (s), 1165 (w), and 1148 (w) cm^−1^. Elemental analysis calcd (%) for C_6_H_4_N_4_O_5_: C 33.97, H 1.90, and N 26.41; found: C 34.01, H 1.92, and N 26.21.

**2-Nitro-1-(nitro-*NNO*-azoxy)benzene (2g)**: Pale yellow crystals, m.p. 38–39 °C (lit. m.p. 39–40 °C) [44], 70% yield. *R*_f_ (petroleum ether/ethyl acetate, 10:1) = 0.55. ^1^H NMR (600.13 MHz, CDCl_3_)*δ*: 7.88–7.92 (m, 3H, H(4), H(5), H(6)), 8.22 (d, 1H, H(3), ^3^*J*_HH_ = 7.3 Hz) ppm. ^13^C NMR (150.90 MHz, CDCl_3_)*δ*: 126.0 (s, C(6)), 126.3 (s, C(3)), 133.8 (s, C(4)), 134.7 (s, C(5)), 137.1 (br. s, C(1)), 142.4 (br. s, C(2)) ppm. The ^1^H–^13^C HSQC and HMBC experiments were used to assign the signals. ^14^N NMR (43.37 MHz, CDCl_3_)*δ*: −20 (NO_2_, ∆ν_½_ = 55 Hz), −39 (N(O)=N–NO_2_, ∆ν_½_ = 15 Hz), −51 (N(O)=N–NO_2_, ∆ν_½_ = 50 Hz) ppm. IR (KBr): ν = 2918 (w), 2887 (w), 1621 (s), 1542 (s), 1494 (s), 1449 (w), 1437 (w), 1406 (w), 1351 (s), 1313 (w), 1280 (s), 1265 (s), 1169 (w), and 1147 (w) cm^−1^. Elemental analysis calcd (%) for C_6_H_4_N_4_O_5_: C 33.97, H 1.90, and N 26.41; found: C 34.00, H 1.94, and N 26.25.

**1-Methoxy-4-(nitro-*NNO*-azoxy)benzene (2i)**: Pale yellow oil, 83% yield. *R*_f_ (petroleum ether/ethyl acetate, 10:1) = 0.35. An analytical sample was obtained by vacuum distillation at 65 °C (0.75 Torr). ^1^H NMR (600.13 MHz, CDCl_3_)*δ*: 3.93 (s, 3H, OMe), 7.01 (d, 2H, H(2), H(6), ^3^*J*_HH_ = 9.2 Hz), 8.09 (d, 2H, H(3), H(5), ^3^*J*_HH_ = 9.2 Hz) ppm. ^13^C NMR (150.90 MHz, CDCl_3_)*δ*: 56.1 (s, OMe), 114.5 (s, C(2), C(6)), 124.3 (s, C(3), C(5)), 135.3 (br. s, C(4)), 164.6 (s, C(1)) ppm. The ^1^H–^13^C HSQC and HMBC experiments were used to assign the signals. ^14^N NMR (43.4 MHz, CDCl_3_)*δ*: −33 (N(O)=N–NO_2_, ∆ν_½_ = 20 Hz), −46 (N(O)=N–NO_2_, ∆ν_½_ = 200 Hz) ppm. IR (KBr): ν = 2978 (w), 2948 (w), 1594 (s), 1541 (m), 1499 (s), 1471 (s), 1417 (m), 1336 (m), 1316 (m), 1258 (s), 1174 (s), and 1118 (s) cm^−1^. Elemental analysis calcd (%) for C_7_H_7_N_3_O_4_: C 42.65, H 3.58, and N 21.31; found: C 42.70, H 3.64, and N 21.15.

**1-Methoxy-3-(nitro-*NNO*-azoxy)benzene (2j)**: Yellow oil, 79% yield. *R*_f_ (petroleum ether/ethyl acetate, 10:1) = 0.57. ^1^H NMR (600.13 MHz, CDCl_3_)*δ*: 3.88 (s, 3H, OMe), 7.24 (dd, 1H, H(6), ^3^*J*_HH_ = 8.3 Hz, ^4^*J*_HH_ = 2.3 Hz), 7.46 (t, 1H, H(5), *J*_HH_ = 8.3 Hz), 7.57 (s, 1H, H(2)), 7.69 (dd, 1H, H(4), ^3^*J*_HH_ = 8.3 Hz, ^4^*J*_HH_ = 1.8 Hz) ppm. ^13^C NMR (150.9 MHz, CDCl_3_)*δ*: 55.9 (s, OMe), 107.1 (s, C(2)), 114.2 (s, C(4)), 121.0 (s, C(6)), 130.4 (s, C(5)), 143.5 (br. s, C(3)), 160.2 (s, C(1)) ppm. The ^1^H–^13^C HSQC and HMBC experiments were used to assign the signals. ^14^N NMR (43.4 MHz, CDCl_3_)*δ*: −34 (N(O)=N–NO_2_, ∆ν_½_ = 30 Hz), −46 (N(O)=N–NO_2_, ∆ν_½_ = 215 Hz) ppm. IR (KBr): ν = 2970 (w), 2942 (w), 2841 (w), 1612 (s), 1585 (m), 1530 (m), 1502 (s), 1472 (m), 1447 (m), 1384 (w), 1335 (m), 1319 (m), 1290 (m), 1271 (s), 1250 (s), 1185 (w), and 1106 (w) cm^−1^. Elemental analysis calcd (%) for C_7_H_7_N_3_O_4_: C 42.65, H 3.58, and N 21.31; found: C 42.72, H 3.59, and N 21.05.

**1-Methoxy-2-(nitro-*NNO*-azoxy)benzene (2k)**: Yellow oil, 51% yield. *R*_f_ (petroleum ether/ethyl acetate, 10:1) = 0.71. ^1^H NMR (300.13 MHz, CDCl_3_)*δ*: 3.89 (s, 3H, OMe), 7.24 (dd, 1H, H(6), ^3^*J*_HH_ = 8.3 Hz, ^4^*J*_HH_ = 2.5 Hz), 7.46 (t, 1H, H(5), ^3^*J*_HH_ = 8.3 Hz), 7.60 (t, 1H, H(4), ^3^*J*_HH_ = 2.3 Hz), 7.69 (dd, 1H, H(3), ^3^*J*_HH_ = 8.1 Hz, ^4^*J*_HH_ = 2.2 Hz) ppm. ^13^C NMR (75.49 MHz, CDCl_3_)*δ*: 55.9 (s, OMe), 107.2 (s, C(4)), 114.3 (s, C(3)), 121.0 (s, C(6)), 130.4 (s, C(5)), 143.8 (br. s, C(2)), 160.4 (s, C(1)) ppm. The ^1^H–^13^C HSQC and HMBC experiments were used to assign the signals. ^14^N NMR (43.4 MHz, CDCl_3_)*δ*: −34 (N(O)=N–NO_2_, ∆ν_½_ = 20 Hz), −45 (N(O)=N–NO_2_, ∆ν_½_ = 115 Hz) ppm. IR (KBr): ν = 2969 (w), 2941 (w), 2841 (w), 1612 (s), 1503 (s), 1472 (m), 1447 (m), 1384 (w), 1335 (m), 1319 (m), 1290 (m), 1271 (s), 1250 (s), 1185 (w), and 1106 (w) cm^−1^. Elemental analysis calcd (%) for C_7_H_7_N_3_O_4_: C 42.65, H 3.58, and N 21.31; found: C 42.66, H 3.58, and N 21.21.

**4-Fluoro-1-(nitro-*NNO*-azoxy)benzene (2l):** Yellow oil (mixture with 4-fluoro-1-nitrobenzene in 1.5:1 ratio), 36% yield. *R*_f_ (petroleum ether/ethyl acetate, 10:1) = 0.65. ^1^H NMR (600.13 MHz, CDCl_3_)δ = 7.28 (m, 2H, H(2), H(6)), 8.19 (m, 2H, H(3), H(5)) ppm. ^13^C NMR (150.9 MHz, CDCl_3_)δ = 117.0 (d, C(3), C(5), ^2^*J*_CF_ = 23.8 Hz), 125.0 (d, C(2), C(6), ^3^*J*_CF_ = 9.9 Hz), 138.7 (br. s, C(1)) ppm. The ^1^H–^13^C HSQC and HMBC experiments were used to assign the signals. ^14^N NMR (43.37 MHz, CDCl_3_)δ = −35 (N(O)=N–NO_2_, ∆ν_½_ = 30 Hz), −48 (N(O)=N–NO_2_, ∆ν_½_ = 130 Hz) ppm.

**4-Chloro-1-(nitro-*NNO*-azoxy)benzene (2m):** Yellowish oil, 52% yield. *R*_f_ (petroleum ether/ethyl acetate, 30:1) = 0.44. An analytical sample was obtained by two-fold vacuum distillation at 65 °C (0.75 Torr). ^1^H NMR (600.13 MHz, CDCl_3_) δ = 7.57 (d, 2H, H(3), H(5), ^3^*J*_H,H_ = 9.0 Hz), 8.11 (d, 2H, H(2), H(6), ^3^*J*_H,H_ = 9.0 Hz) ppm. ^13^C NMR (150.9 MHz, CDCl_3_) δ = 123.6 (s, C(2), C(6)), 130.0 (s, C(3), C(5)), 141.3 (br. s, C(1), C(4)) ppm. The ^1^H–^13^C HSQC and HMBC experiments were used to assign the signals. ^14^N NMR (43.37 MHz, CDCl_3_) δ = −36 (N(O)=N–NO_2_, ∆ν_½_ = 35 Hz), −48 (N(O)=N–NO_2_, ∆ν_½_ = 135 Hz) ppm. IR (KBr): ν = 2873 (w), 1610 (s), 1584 (m), 1523 (w), 1478 (s), 1401 (w), 1344 (w), 1315 (m), 1269 (s), 1174 (m), 1139 (w), 1123 (w), and 1108 (w) cm^−1^. Elemental analysis calcd (%) for C_6_H_4_ClN_3_O_3_: C 35.75, H 2.00, and N 20.85; found: C 35.79, H 2.03, and N 20.63.

**2,4-Dichloro-1-(nitro-*NNO*-azoxy)benzene (2n):** Brown crystals, m.p. 45–46 °C, 53% yield. *R*_f_ (petroleum ether/ethyl acetate, 10:1) = 0.50. An analytical sample was obtained by recrystallization from hexane at −20 °C. ^1^H NMR (500.13 MHz, CDCl_3_) δ = 7.47 (d, 1H, H(5), ^3^*J*_H,H_ = 8.6 Hz), 7.62 (s, 1H, H(3)), 7.71 (d, 1H, H(6), ^3^*J*_H,H_ = 8.6 Hz) ppm. ^13^C NMR (125.76 MHz, CDCl_3_) δ = 126.5 (s, C(6)), 128.4 (s, C(5)), 131.7 (s, C(3)), 139.5 (s, C(2)), 140.0 (br.s, C(1)) ppm. The ^1^H–^13^C HSQC and HMBC experiments were used to assign the signals. ^14^N NMR (36.14 MHz, CDCl_3_) δ = −36 (N(O)=N–NO_2_, ∆ν_½_ = 25 Hz), −49 (N(O)=N–NO_2_, ∆ν_½_ = 100 Hz) ppm. IR (KBr): ν = 2940 (w), 2875 (w), 1619 (s), 1582 (s), 1566 (s), 1482 (s), 1464 (s), 1393 (w), 1376 (m), 1342 (w), 1317 (m), 1274 (s), 1145 (m), and 1107 (m) cm^−1^. Elemental analysis calcd (%) for C_6_H_3_Cl_2_N_3_O_3_: C 30.54, H 1.28, and N 17.80; found: C 30.55, H 1.31, and N 17.63.

**2,6-Dichloro-1-(nitro-*NNO*-azoxy)benzene (2o):** Brown crystals, m.p. 53–54 °C (lit. m.p. 54–56 °C) [44], 64% yield. *R*_f_ (petroleum ether/ethyl acetate, 10:1) = 0.57. ^1^H NMR (500.13 MHz, CDCl_3_) δ = 7.80–7.93 (m, 3H, H(3), H(4), H(5)) ppm. ^13^C NMR (125.76 MHz, CDCl_3_) δ = 128.4 (s, C(2), C(6)), 130.6 (s, C(3), C(5)), 135.1 (s, C(4)), 140.0 (br.s, C(1)) ppm. The ^1^H–^13^C HSQC and HMBC experiments were used to assign the signals. ^14^N NMR (36.14 MHz, CDCl_3_) δ = −37 (N(O)=N–NO_2_, ∆ν_½_ = 25 Hz), −53 (N(O)=N–NO_2_, ∆ν_½_ = 85 Hz) ppm. IR (KBr): ν = 2917 (w), 2892 (w), 1624 (s), 1578 (m), 1556 (m), 1492 (s), 1448 (m), 1388 (w), 1370 (w), 1330 (m), 1304 (s), 1277 (s), 1206 (m), and 1168 (m) cm^−1^. Elemental analysis calcd (%) for C_6_H_3_Cl_2_N_3_O_3_: C 30.54, H 1.28, and N 17.80; found: C 30.60, H 1.35, and N 17.53.

**2,4,6-Trichloro-1-(nitro-*NNO*-azoxy)benzene (2p):** Beige crystals, m.p. 80–81 °C (lit. m.p. 80–81 °C) [45], 80% yield. *R*_f_ (petroleum ether/ethyl acetate, 40:1) = 0.70. ^1^H NMR (500.13 MHz, CDCl_3_) δ = 7.84 (s, 2H, H(3), H(5)) ppm. ^13^C NMR (125.76 MHz, CDCl_3_) δ = 129.5 (br. s, C(4)), 130.8 (s, C(3), C(5)), 135.1 (s, C(2), C(6)), 140.3 (br.s, C(1)) ppm. The ^1^H–^13^C HSQC and HMBC experiments were used to assign the signals. ^14^N NMR (36.14 MHz, CDCl_3_) δ = −36 (N(O)=N–NO_2_, ∆ν_½_ = 30 Hz), −52 (N(O)=N–NO_2_, ∆ν_½_ = 90 Hz) ppm. IR (KBr): ν = 2952 (w), 2926 (w), 2901 (w), 2855 (w), 1624 (s), 1566 (s), 1494 (s), 1451 (m), 1430 (m), 1390 (m), 1371 (m), 1330 (w), 1286 (s), 1196 (w), and 1128 (m) cm^−1^. Elemental analysis calcd (%) for C_6_H_2_Cl_3_N_3_O_3_: C 26.65, H 0.75, and N 15.54; found: C 26.76, H 0.77, and N 15.32.

**4-Bromo-1-(nitro-*NNO*-azoxy)benzene (2q):** Yellowish crystals, m.p. 74–75 °C (lit. m.p. 74–76 °C) [44], 50% yield. *R*_f_ (petroleum ether/ethyl acetate, 40:1) = 0.42. An analytical sample was obtained by three-fold recrystallization from mixture hexane/CH_2_Cl_2_ (10:1) at −20 °C. ^1^H NMR (600.13 MHz, CDCl_3_) δ = 7.73 (d, 2H, H(3), H(5), ^3^*J*_H,H_ = 8.9 Hz), 8.01 (d, 2H, H(2), H(6), ^3^*J*_H,H_ = 8.9 Hz) ppm. ^13^C NMR (150.9 MHz, CDCl_3_) δ = 123.7 (s, C(2), C(6)), 129.8 (br. s, C(4)), 133.0 (s, C(3), C(5)), 141.7 (br. s, C(1)) ppm. The ^1^H–^13^C HSQC and HMBC experiments were used to assign the signals. ^14^N NMR (43.37 MHz, CDCl_3_) δ = −36 (N(O)=N–NO_2_, ∆ν_½_ = 20 Hz), −48 (N(O)=N–NO_2_, 75 Hz) ppm. IR (KBr): ν = 2921 (w), 2870 (w), 1606 (s), 1576 (m), 1519 (w), 1473 (s), 1396 (w), 1310 (w), 1280 (m), 1175 (w), and 1108 (w) cm^−1^. Elemental analysis calcd (%) for C_6_H_4_BrN_3_O_3_: C 29.29, H 1.64, and N 17.08; found: C 29.31, H 1.65, and N 16.91.

**2,4-Dibromo-1-(nitro-*NNO*-azoxy)benzene (2r):** Orange crystals, m.p. 71–72 °C, 34% yield. *R*_f_ (petroleum ether/ethyl acetate, 40:1) = 0.54. ^1^H NMR (600.13 MHz, CDCl_3_) δ = 7.60 (d, 1H, H(6), ^3^*J*_H,H_ = 8.5 Hz), 7.67 (d, 1H, H(5), ^3^*J*_H,H_ = 8.4 Hz), 7.96 (s, 1H, H(3)) ppm. ^13^C NMR (150.9 MHz, CDCl_3_) δ = 117.4 (s, C(4) or C(2)), 127.3 (s, C(6)), 128.1 (s, C(2) or C(4)), 132.6 (s, C(5)), 138.2 (s, C(3)), 143.0 (br. s, C(1)) ppm. The ^1^H–^13^C HSQC and HMBC experiments were used to assign the signals. ^14^N NMR (43.37 MHz, CDCl_3_) δ = −37 (N(O)=N–NO_2_, ∆ν_½_ = 35 Γц), −47 (N(O)=N–NO_2_, ∆ν_½_ = 80 Γц) ppm. IR (KBr): ν = 2931 (w), 2881 (w), 2811 (w), 1606 (s), 1568 (s), 1535 (s), 1479 (s), 1459 (s), 1387 (w), 1371 (m), 1331 (m), 1288 (s), 1245 (s), 1149 (m), and 1109 (w) cm^−1^. Elemental analysis calcd (%) for C_6_H_3_Br_2_N_3_O_3_: C 22.18, H 0.93, and N 12.93; found: C 22.18, H 0.94, and N 12.71.

**2,6-Dibromo-1-(nitro-*NNO*-azoxy)benzene (2s):** Beige crystals, m.p. 97–98 °C, 53% yield. *R*_f_ (petroleum ether/ethyl acetate, 6:1) = 0.49. ^1^H NMR (600.13 MHz, CDCl_3_) δ = 7.35 (t, ^3^*J*_H,H_ = 8.1 Hz, 1H, H(4)), 7.70 (d, ^3^*J*_H,H_ = 8.2 Hz, 2H, H(3), H(5)) ppm. ^13^C NMR (150.9 MHz, CDCl_3_) δ = 118.1 (s, C(2), C(6)), 133.8 (s, C(3), C(5)), 134.0 (s, C(4)), 143.7 (br. s, C(1)) ppm. The ^1^H–^13^C HSQC and HMBC experiments were used to assign the signals. ^14^N NMR (43.37 MHz, CDCl_3_) δ = −37 (N(O)=N–NO_2_, ∆ν_½_ = 25 Hz), −48 (N(O)=N–NO_2_, ∆ν_½_ = 125 Hz) ppm. IR (KBr): ν = 2917 (w), 2889 (w), 2849 (w), 1633 (s), 1567 (m), 1535 (s), 1490 (s), 1441 (m), 1422 (w), 1370 (w), 1327 (w), 1295 (m), 1274 (s), 1288 (s), 1201 (m), and 1161 (w) cm^−1^. Elemental analysis calcd (%) for C_6_H_3_Br_2_N_3_O_3_: C 22.18, H 0.93, and N 12.93; found: C 22.18, H 0.94, and N 12.71.

**2,4,6-Tribromo-1-(nitro-*NNO*-azoxy)benzene (2t):** Beige crystals, m.p. 113–114 °C, 53% yield. *R*_f_ (petroleum ether/ethyl acetate, 40:1) = 0.46. ^1^H NMR (500.13 MHz, CDCl_3_) δ = 7.87 (s, 2H, H(3), H(5)) ppm. ^13^C NMR (125.76 MHz, CDCl_3_) δ = 117.4 (s, C(2), C(6)), 126.1 (s, C(4)), 134.8 (s, C(3), C(5)), 141.6 (s, C(4)) ppm. The ^1^H–^13^C HSQC and HMBC experiments were used to assign the signals. ^14^N NMR (43.37 MHz, CDCl_3_) δ = −38 (N(O)=N–NO_2_, ∆ν_½_ = 25 Hz), −50 (N(O)=N–NO_2_, ∆ν_½_ = 115 Hz) ppm. IR (KBr): ν = 2891 (w), 2814 (w), 1620 (s), 1551 (m), 1493 (m), 1430 (w), 1415 (w), 1370 (w), 1350 (w), 1329 (w), 1282 (m), 1248 (w), 1198 (w), 1161 (w), 1131 (w), and 1112 (w) cm^−1^. Elemental analysis calcd (%) for C_6_H_2_Br_3_N_3_O_3_: C 17.85, H 0.50, and N 10.41; found: C 17.87, H 0.52, and N 10.20.


**Procedure for Gram Scale Electrochemical Synthesis of (Nitro-*NNO*-Azoxy)benzene (2a) (Experimental details for Figure 3).**


An undivided 200 mL jacketed three-necked electrochemical cell was equipped with a cylindrical carbon felt anode (90 × 55 mm^2^, S = 49.5 cm^2^) and a platinum wire cathode placed inside the anode space, and connected to a DC regulated power supply. A solution of nitrosobenzene **1a** (10.0 mmol, 1.07 g) and ADN (4.0 equiv., 40.0 mmol, 4.96 g) in MeCN (200 mL) was electrolyzed using constant current conditions (*I* = 660 mA) employing water-jet cooling (water temperature ca. 20 °C) to prevent reaction heating. After passing 2.0 F∙mol^−1^ of electricity (reaction time 49 min), electrodes were washed with CH_2_Cl_2_ (3 × 100 mL). The combined organic phase was washed with H_2_O (200 mL) and brine (200 mL), dried over Na_2_SO_4_, and solvent removed in vacuo. Product **2a** (1.086 g, 6.5 mmol, 65%) was isolated by column chromatography on silica gel (*R*_f_ = 0.51, petroleum ether/ethyl acetate, 10:1).


**Reaction in Divided Electrochemical Cell (Experimental details for Figure 4A)**


A divided H-type electrochemical cell (volume of each compartment—30 mL, divided with DuPont Nafion^®^ N-117 membrane) was equipped with a carbon felt anode (30 × 15 mm^2^) and a platinum plate cathode (30 × 15 mm^2^), and connected to a DC regulated power supply. Electrodes were completely immersed in the solution, giving S = 4.5 cm^2^ of working surface. A solution of nitrosobenzene **1a** (1.0 mmol, 107 mg) and ADN (4.0 equiv., 4.0 mmol, 496 mg) in MeCN (20 mL) was placed in the anodic compartment of the cell and a solution of ADN (4.0 mmol, 496 mg) in MeCN (20 mL) was placed in the cathodic compartment of the cell. Solutions were electrolyzed using constant current conditions (*I* = 60 mA) at 23–25 °C under magnetic stirring. After passing 2.0 F of electricity per mole of **1a** (reaction time 54 min), electrodes were washed with CH_2_Cl_2_ (2 × 20 mL). The organic phases from anodic and cathodic compartments were separately evaporated under a water-jet vacuum. The yield of **2a** was determined according to ^1^H NMR spectroscopy using 2-methyl-2-nitropropane as an internal standard and checked by ^14^N NMR.


**Potential Monitoring During Electrolysis (Experimental details for Figure 4B)**


An undivided 20 mL jacketed electrochemical cell was equipped with a carbon felt anode (30 × 15 mm^2^), a platinum wire cathode, and a reference Ag/AgNO_3_ electrode linked to the solution by a porous glass diaphragm, and connected to a computer-assisted potentiostat. Electrodes were completely immersed in the solution, giving S = 4.5 cm^2^ of working anode surface. A mixture of nitrosobenzene **1a** (1.0 mmol, 107 mg) and ADN (4.0 equiv., 4.0 mmol, 496 mg) in MeCN (20 mL) was electrolyzed using constant current conditions (*I* = 60 mA) at 23–25 °C under magnetic stirring. The observed electrochemical potential at anode was recorded as the reaction proceeded.


**Reaction under Constant Potential Electrolysis (Experimental details for Figure 4C)**


An undivided 20 mL jacketed electrochemical cell was equipped with a carbon felt anode (30 × 15 mm^2^), a platinum wire cathode, and a reference Ag/AgNO_3_ electrode linked to the solution by a porous glass diaphragm, and connected to a computer-assisted potentiostat. Electrodes were completely immersed in the solution, giving S = 4.5 cm^2^ of working anode surface. A mixture of nitrosobenzene **1a** (1.0 mmol, 107 mg) and ADN (4.0 equiv., 4.0 mmol, 496 mg) in MeCN (20 mL) was electrolyzed using constant potential conditions (*E_cell_* = 1.8 V vs. Ag/AgNO_3_) at 23–25 °C under magnetic stirring. After passing 2.0 or 5.5 F of electricity per mole of **1a** (reaction time was 1 h and 3 h, respectively), electrodes were washed with CH_2_Cl_2_ (3 × 20 mL). The combined organic phase was washed with H_2_O (20 mL) and brine (20 mL), dried over Na_2_SO_4_, and the solvent removed in vacuo. The yield of **2a** was determined according to ^1^H NMR spectroscopy using 2-methyl-2-nitropropane as an internal standard and checked by ^14^N NMR.


**Study of ADN Constant Current Electrolysis in an Undivided and Divided Electrochemical Cell (Experimental details for Figure 4D)**


(a)An undivided 20 mL jacketed electrochemical cell was equipped with a carbon felt anode (30 × 15 mm^2^) and a platinum wire cathode, and connected to a DC regulated power supply. Electrodes were completely immersed in the solution, giving S = 4.5 cm^2^ of working anode surface. A solution ADN (4.0 mmol, 496 mg) in MeCN (20 mL) was electrolyzed using constant current conditions (*I* = 60 mA) at 23–25 °C under magnetic stirring. After passing 2 × 10^−3^ F of electricity (reaction time 54 min), electrodes were washed with CH_2_Cl_2_ (3 × 20 mL). The solvent was removed in vacuo, and the resulting reaction mixture was analysed by ^14^N spectroscopy in D_2_O.(b)A divided H-type electrochemical cell (volume of each compartment—30 mL, divided with DuPont Nafion^®^ N-117 membrane) was equipped with a carbon felt anode (30 × 15 mm^2^) and a platinum plate cathode (30 × 15 mm^2^), and connected to a DC regulated power supply. Electrodes were completely immersed in the solution, giving S = 4.5 cm^2^ of working surface. Solutions of ADN (4.0 mmol, 496 mg) in MeCN (20 mL) were placed in the anodic and cathodic compartments of the cell. Solutions were electrolyzed using constant current conditions (*I* = 60 mA) at 23–25 °C under magnetic stirring. After passing 2 × 10^−3^ F of electricity (reaction time 54 min), electrodes were washed with CH_2_Cl_2_ (2 × 20 mL). The organic phases from the anodic and cathodic compartments were separately evaporated under water-jet vacuum. The resulting reaction mixtures were analysed by ^14^N spectroscopy in D_2_O.


**Cyclic Voltammetry Studies**


Cyclic voltammetry (CV) was implemented using an «Econix» computer-assisted potentiostat IPC-Pro M (scan rate 100 mV∙s^−1^; potential setting precision 0.25 mV; scan rate error 1.0%) in a 10 mL water-jacketed five-neck glass conic electrochemical cell. CV curves were recorded employing a three-electrode scheme. In a typical case, 5 mL of a solution was utilized. A disc glassy-carbon electrode (d = 3 mm) served as the working electrode. A platinum wire served as the auxiliary electrode. An Ag/AgNO_3_ (0.1 M) in 0.1 M *n*-Bu_4_NBF_4_/MeCN electrode was used as the reference electrode and was linked to the solution by a porous glass diaphragm. The solutions were kept under thermally controlled conditions at 21 ± 0.5 °C and deaerated by bubbling argon. Electrochemical experiments were performed under an argon atmosphere. The working electrode was polished before recording each CV curve.


**Fungicidal activity tests (experimental details for Table 2).**


The strains used in this work were obtained from the collection of the All-Russian Research Institute for Phytopathology (B. Vyazemy, Moscow reg., Russia).

Six phytopathogenic fungi from different taxonomic classes were utilized in fungicidal activity measurements (*V.i.*—Venturia inaequalis MRA-16-2, *R.s.*—Rhizoctonia solani 100063, *F.o.*—Fusarium oxysporum FO-8, *F.m.*—Fusarium moniliforme 100,146, *B.s*.—Bipolaris sorokiniana MRB(V)-1, *S.s*.—Sclerotinia sclerotiorum 100,033) using the standard poison food technique [13,98,99,100,101,102]. Tested substances dissolved in acetone (1 mg/mL) were mixed with liquid sugar-potato agar at 50–55 °C to get a final concentration of 30 mg/L. The mixtures of agar and test substances were poured into sterile Petri dishes and allowed to cool to room temperature. Mycelial pieces from the peripheral growth zone of 3–5-day-old fungal cultures were transferred to the test dishes using a needle. Colonies grown in medium mixed with acetone without a substance served as controls. Diameters of the fungal colonies were measured after 72 h. Each experiment was repeated three times, and tests with *V. inaequalis* were repeated five times. Mycelial growth suppression was calculated by the formula ((D_c_ − D_s_)/D_c_) × 100%, where D_c_ is the average control colony diameter and D_s_ is the average colony diameter in the presence of the tested substance.

## 5. Conclusions

In summary, we have disclosed the electrochemical coupling of nitrosoarenes with ammonium dinitramide leading to substituted (nitro-*NNO*-azoxy)benzenes. Dinitramide salt plays the role of both reactant and electrolyte. The reaction is operationally convenient and proceeds in an undivided cell under constant current conditions. Moreover, high current densities are achievable without significant loss in selectivity, which allows for efficient scaling up of the synthesis. Presumably, the excess of ammonium dinitramide is crucial for the suppression of the cathodic reduction of the target product. Compared to known approaches to nitro-*NNO*-azoxy compounds, the developed electrochemical method consists of a single step, does not require expensive and hazardous nitronium salts, and is effective for the synthesis of (nitro-*NNO*-azoxy)arenes with various substituents.

The synthesized nitro-*NNO*-azoxy compounds were discovered as potent fungicides with in vitro activity against a broad range of phytopathogenic fungi comparable to the activity of commercial fungicides (triadimefon). The proposed simple and scalable method for the synthesis of fungicidal nitro-*NNO*-azoxy compounds makes them a promising starting point for the development of a novel and easily available class of fungicides.

## Data Availability

Data is contained within the article or Appendix A.

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
