# Peer review of "Ammonium Dinitramide as a Prospective N–NO_2_ Synthon: Electrochemical Synthesis of Nitro-*NNO*-Azoxy Compounds from Nitrosoarenes"

_molecules, 2024, doi:10.3390/molecules29235563_

Round 1
Reviewer 1 Report
Comments and Suggestions for Authors
In this work, the authors have disclosed electrochemical coupling of nitrosoarenes with ammonium dinitramide leading to substituted (nitro-NNO-azoxy)benzenes. Dinitramide salt plays the role of both reactant and electrolyte. The reaction is operationally convenient and proceeds in an undivided cell under constant current conditions. This work well demonstrates that the electrochemical method is an efficient method to produce the specific organic compounds with high selectivity. The authors well wrote this work and analyzed the experimental tests. However, there are some issues to be addressed before publish in Molecules.
Detailed comments:
1. The words of I-T curves in Scheme 4, Figure 1 and Figure 2 are too small to clear observed, and the authors should modify this issue.
2. How about the impedances of the different reactants in electrolyte? Whether the impedance is the main factor to influence the selectivity of different reactants.
3. Catalytic activity is closely related with the surface area of catalysts exposed in electrolyte (Adv. Sci. 2018, 5, 1800036). Generally, the electrocatalysts possess the different electrochemical active surface areas (ECSA) in the various electrolytes featured by different reactants, indicating the different exposed catalytic sites in every reaction. Therefore, the authors should provide the ECSA data of the Pt and C catalysts in various reactants, and the reference should be cited in this part (Adv. Sci. 2018, 5, 1800036).
Author Response
Reviewer’s comment: In this work, the authors have disclosed electrochemical coupling of nitrosoarenes with ammonium dinitramide leading to substituted (nitro-NNO-azoxy)benzenes. Dinitramide salt plays the role of both reactant and electrolyte. The reaction is operationally convenient and proceeds in an undivided cell under constant current conditions. This work well demonstrates that the electrochemical method is an efficient method to produce the specific organic compounds with high selectivity. The authors well wrote this work and analyzed the experimental tests. However, there are some issues to be addressed before publish in Molecules.
Answer: Thank you for reading and positive assessment of our manuscript.
Reviewer’s comment: 1. The words of I-T curves in Scheme 4, Figure 1 and Figure 2 are too small to clear observed, and the authors should modify this issue.
Answer: The captions in Scheme 4, Figure 1, and Figure 2 were corrected.
Reviewer’s comment: 2. How about the impedances of the different reactants in electrolyte? Whether the impedance is the main factor to influence the selectivity of different reactants.
Answer: Thank you for this idea. Electrochemical impedance spectroscopy is a powerful tool in many areas, but it is still not so widely known and used in electrochemical organic synthesis to investigate mechanisms of electrode processes. We have just begun impedance studies for the discovered process and unambiguous interpretation of the results demands additional experiments. We are planning separate paper devoted to the in-depth mechanistic studies supported by electronic structure calculations. Introduction of such results in the present paper is not possible due to time limitations. Moreover, the main goal of the present paper is to show practical applicability of the discovered reaction and we wanted to focus on main practical achievements. Voltammetry results presented in figure 1 indicate that different mechanisms can take place for electron-rich and electron-poor nitrosoarenes, which have significantly different oxidation potentials but successfully give target products. Apparently, the key feature responsible for high selectivity is the usage of AND as electrolyte in excess amount relative to nitrosoarene (for example, see Tbale 1, entry 1 vs. entry 2; entry 3) to force ADN participation in anodic oxidation and suppress side oxidation processes of nitrosoarene itself. The plausible role of Pt as cathode material is to catalyze hydrogen evolution (see Scheme 5) instead of side processes of reduction of the target product.
Reviewer’s comment: 3. Catalytic activity is closely related with the surface area of catalysts exposed in electrolyte (Adv. Sci. 2018, 5, 1800036). Generally, the electrocatalysts possess the different electrochemical active surface areas (ECSA) in the various electrolytes featured by different reactants, indicating the different exposed catalytic sites in every reaction. Therefore, the authors should provide the ECSA data of the Pt and C catalysts in various reactants, and the reference should be cited in this part (Adv. Sci. 2018, 5, 1800036).
Answer: Thank you for this essential comment. All electrodes except for carbon felt (CF) electrodes were polished flat plates with real surface areas the same as geometrical surface areas. We have added specification of the used commercial carbon felt and its estimated real area from cyclic voltammetry measurements. The mentioned paper was cited.
Added text: “All electrodes except for carbon felt (CF) electrodes were polished flat plates with equal real surface areas and geometrical surface areas. Commercial carbon felt (PANCF3200300, produced from polyacrylonitrile, carbon content ≥98%, 3 mm thick-ness) was used as is. Electrode surface area is one of the key factors affecting the selectivity of electrode processes [97] due to dependence of electrochemical potential on current density and other factors. Carbon felt electrode surface was estimated by the integration of cyclic voltammograms (15 cycles were done to achieve constant voltammogram) of ferrocene (0.001 M) in MeCN containing n-Bu4NBF4 (0.1 M) as sup-porting electrolyte for carbon felt sample and glassy carbon plate with known surface area as working electrodes. Specific surface area of carbon felt sample estimated this way was about 0.05 m2·g-1 (carbon felt density was 0.14 g·cm-3).”
Reviewer 2 Report
Comments and Suggestions for Authors
The authors presented a manuscript in which they successfully established a methodology for electrochemical synthesis of nitro-NNO-azoxy compounds starting from nitroarenes using popular racket propellant as an active compound. The work presents a synthesis optimization effort, the scope of substrates, a scale-up procedure, and anti-fungal properties tests. The newly obtained compounds are minutely characterized, and procedures are scrupulously described. I belive that the manuscript is worth publishing after minor, editoral correction to the manuscript:
- I suggest to redo the Table 1 so it was more readable. For example withering the spaces in between entries might help. As now it is very hard to read.
- the introduction contains a lot of references; some are repetitive in content. I suggest cutting out about 20.
Author Response
Reviewer’s comment: The authors presented a manuscript in which they successfully established a methodology for electrochemical synthesis of nitro-NNO-azoxy compounds starting from nitroarenes using popular racket propellant as an active compound. The work presents a synthesis optimization effort, the scope of substrates, a scale-up procedure, and anti-fungal properties tests. The newly obtained compounds are minutely characterized, and procedures are scrupulously described.
Answer: We thank the Reviewer for positive assignment of our manuscript.
Reviewer’s comment: I belive that the manuscript is worth publishing after minor, editoral correction to the manuscript:
– I suggest to redo the Table 1 so it was more readable. For example withering the spaces in between entries might help. As now it is very hard to read.
Answer: Thank you for this note, Table 1 was corrected: spaces were added to values separated by “/” sign, repeating condition variation (“I = 15 mA, (+)Pt / Ptw (–)”) was moved to note «c» for clearer view of the table.
Reviewer’s comment: The introduction contains a lot of references; some are repetitive in content. I suggest cutting out about 20.
Answer: According to the Reviewer’s advice, the number of cited references was reduced. It should be noted that other Reviewers suggested additional citations, which were introduced. The following sentence was removed along with corresponding citations:
Removed text: Great attention is being paid to the development of electrochemical oxidative coupling [26] with the formation of C–C [27–29], C–O [30–33], C–N [34–37], C–S [38–41] and C–P [42–45] bonds, CH- and C=C-functionalization [46,47], and heterocycle construction [48,49] approaches.
Reviewer 3 Report
Comments and Suggestions for Authors
The manuscript by Klenov and co-authors presents work on the electrochemical synthesis of nitro-NNO azoxy derivatives from nitroarenes, using ammonium dinitramide as an N-NO2 synthon. The article demonstrates a robust scientific approach toward electrochemistry. Recently, electrochemical methods have been applied to various syntheses of important molecules, leveraging the use of current as a reagent to promote efficient and elegant reactions. Here, the authors apply this approach to synthesize N-N compounds.
Based on the references these compounds are quite important for both materials and in medicinal chemistry.
The paper is good for several reasons: 1) it addresses N-N bond formation, 2) it develops a new N-NO2 synthon, and 3) it utilizes an electrochemical method for this purpose. However, one drawback of the method could be the use of nitrosoarenes as the phenyl-N source due to their high toxicity. Unfortunately, the authors did not provide information on how they handled this toxic chemical in the materials and methods or in the ESI, which would have been useful for readers.
The authors have prepared a thorough and informative ESI, beneficial for readers.
They should include additional references to useful energetic materials rather than relying on self-citations.
In the abstract, the authors mention that known methods use nitronium salts, which are hazardous; however, they also use nitrosoarenes, which are comparably toxic. The abstract should be revised to address these toxicity concerns more clearly.
The English language should be improved throughout the manuscript.
Did the authors observe any dimerization of nitrosobenzenes under the optimized conditions, which could also lead to N-N bond formation?
What will happen if the air is removed and flushed with nitrogen inside the reaction chamber. Does oxygen in the air have any role in this reaction?
The manuscript can be considered for publication based on the development of a new electrochemical method, the achievement of N-N bond formation, the synthesis of important N-NO2 azoxy derivatives, and the evaluation of fungicidal activity for the synthesized compounds.
Comments on the Quality of English Language
The English language is fine but need improvement.
Author Response
Reviewer’s comment: The manuscript by Klenov and co-authors presents work on the electrochemical synthesis of nitro-NNO azoxy derivatives from nitroarenes, using ammonium dinitramide as an N-NO2 synthon. The article demonstrates a robust scientific approach toward electrochemistry. Recently, electrochemical methods have been applied to various syntheses of important molecules, leveraging the use of current as a reagent to promote efficient and elegant reactions. Here, the authors apply this approach to synthesize N-N compounds.
Based on the references these compounds are quite important for both materials and in medicinal chemistry.
The paper is good for several reasons: 1) it addresses N-N bond formation, 2) it develops a new N-NO2 synthon, and 3) it utilizes an electrochemical method for this purpose.
The authors have prepared a thorough and informative ESI, beneficial for readers.
Answer: We thank the Reviewer for positive assignment of our manuscript.
Reviewer’s comment: However, one drawback of the method could be the use of nitrosoarenes as the phenyl-N source due to their high toxicity. Unfortunately, the authors did not provide information on how they handled this toxic chemical in the materials and methods or in the ESI, which would have been useful for readers.
Answer: All practical syntheses of nitro-NNO-azoxy arenes are based on nitrosoarenes (Scheme 1), thus, they still can not be avoided. Nitrosoarenes containing C-N=O fragment are widely used in organic synthesis and as a rule they are much less dangerous compared to N-nitroso compounds. However, we are thankful to the Reviewer for this important note. Corresponding text was added to the beginning of experimental part:
Added text: “Caution! Dinitramide salts should be treated carefully as shock sensitive explosive substances, which stability can depend on preparation method and sample quality, mixtures with reducing agents can be explosive. Nitrosoarenes must be handled with care as potentially toxic and carcinogenic compounds.”
Reviewer’s comment: They should include additional references to useful energetic materials rather than relying on self-citations.
Answer: According to the Reviewer’s advice we have rephrased the corresponding sentence and added references about useful energetic materials.
Old text: “In recent years, this unique class of nitrogen-oxygen compounds has attracted considerable attention as potential energetic materials with outstanding performance.[70–79]”
Corrected version: “In recent years, nitro-NNO-azoxy group has attracted considerable attention as useful fragment in the design of potential energetic materials with outstanding performance [47–56], some of which are among the most energetic compounds, such as TKX-50 [57], nitroazole fused 1,2,3,4-tetrazines [58,59], N-azo nitro-1,2,3-triazoles [60], dinitro-pyrazolo-triazine (PTX) [61], bi-1,2,4-triazole with trinitromethyl groups [62], azido and tetrazolo 1,2,4,5-tetrazine N-oxides [63].”
Reviewer’s comment: In the abstract, the authors mention that known methods use nitronium salts, which are hazardous; however, they also use nitrosoarenes, which are comparably toxic. The abstract should be revised to address these toxicity concerns more clearly.
Answer: We agree with the Reviewer, that nitrosoarenes are toxic in terms of inhalation or skin irritation. However, they are not avoided in alternative (previously known methods) and do not require special handling conditions. Opposite, nitronium salts, especially nitronium tetrafluoroborate or perchlorate, are very hygroscopic compounds, which upon hydrolysis releases corrosive acids (HF, HClO4, HNO3), that can cause skin burns. In addition, nitronium perchlorate is a strong oxidant and requires additional handling and technique conditions, as it can be easily detonated.
The following sentence in the abstract section was corrected:
Old text: “Compared to known approaches to nitro-NNO-azoxy compounds involving two chemical steps (azocoupling, nitration) and demanding expensive and hazardous nitronium salts…”
Corrected version: “Compared to known approaches to nitro-NNO-azoxy compounds involving two chemical steps (azocoupling, nitration) and demanding expensive, corrosive and hygroscopic nitronium salts…”
Reviewer’s comment: The English language should be improved throughout the manuscript.
Answer: According to the Reviewer’s advice, additional language polishing was made.
Reviewer’s comment: Did the authors observe any dimerization of nitrosobenzenes under the optimized conditions, which could also lead to N-N bond formation?
Answer: Thank you for this question. Dimerization was not observed under optimized conditions. The major observed competitive product in discovered nitro-NNO-azoxylation is nitrobenzene, derived from direct oxidation of nitrosobenzene. Reversible dimerization of nitrosoarenes is favored by ortho-substitution and disfavored by electron-donor substituents in para-position of nitrosoarene [Chem. Rev. 2016, 116, 1, 258, DOI: 10.1021/cr500520s], and we did not observe any correlation between yield of the target product and reversible dimerization behavior. Another possible process related to dimerization is the formation of azoxyarenes from nitrosoarenes (“reductive dimerizaion”). During optimization studies we indeed observed noticeable amounts of azoxybenzene, especially while DMF, MeOH or MeCN/H2O mixture were used as solvents. The latter correlates with previously reported results (ChemElectroChem 2024, 11, e202300486, DOI:10.1002/celc.202300486), which stated that nitrosobenzenes undergo reductive dimerization on the cathode surface in the presence of water traces.
Reviewer’s comment: What will happen if the air is removed and flushed with nitrogen inside the reaction chamber. Does oxygen in the air have any role in this reaction?
Answer: We carried out the reaction under an argon atmosphere (see Table S1, run 6). Negligible change in the yield of 2a (compared to Table S1, run 4) let us conclude that the presence of air does not affect the reaction result.
Reviewer’s comment: The manuscript can be considered for publication based on the development of a new electrochemical method, the achievement of N-N bond formation, the synthesis of important N-NO2 azoxy derivatives, and the evaluation of fungicidal activity for the synthesized compounds.
Answer: Thank you for the high evaluation of our work.
Reviewer 4 Report
Comments and Suggestions for Authors
This is a thorough report on a novel synthetic methodology. I recommend that this article is accepted after minor revisions, mostly related to the language of the paper. Only a few minor comments on the science.
"Solvents were purified before use, according to standard procedures." (line 273). Specifically say how you purified solvents, this will help with the reproducibility.
I am a little unclear how you sublimed an oil. "An analytical sample was obtained by sublimation at...) (line 386 and other locations).
Consider depositing your crystal structures to the CCDC and providing CCDC numbers for readers.
Comments on the Quality of English Language
In the abstract make sure results are presented clearly. You shouldn't be talking about "the proposed electrochemical approach (line 17)" you should be stating your results. Also the phrase: "which guarantees outstanding reproducibility" (line 22) is awkward. You should just state a key result that demonstrates your newly developed procedure is scalable.
References need to be reformatted to be before commas, periods, etc.
Do not use the word etc. (line 34)
Do not use the phrase "by one of the authors of this paper" (line 73). Say something like Churakov et al.
Use previously reported instead of known previously (line 143). You don't have to state that "all others were synthesized for the first time" this should be implied.
Don't say Quite interesting results were obtained from the experiment with ad divided electrochemical cell" (line 214). You can't define results as interesting.
In the second part of our study synthesized.....were discovered as a new class of fungicides. (line 236-237) This is phrased awkwardly.
rotary evaporated (line 323) is not correct language. Solvent removed in vacuo.
Author Response
Reviewer’s comment: This is a thorough report on a novel synthetic methodology. I recommend that this article is accepted after minor revisions, mostly related to the language of the paper. Only a few minor comments on the science.
Answer: We thank the Reviewer for positive assignment of our manuscript.
Reviewer’s comment: – Solvents were purified before use, according to standard procedures." (line 273). Specifically say how you purified solvents, this will help with the reproducibility.
Answer: We thank the Reviver for this comment. The purification of solvent was described in more detail.
Old text: “Solvents were purified before use, according to standard procedures.”
Corrected version: “Solvents were purified before use, according to standard procedures. MeCN was distilled over P2O5 and then over CaH2, acetone was distilled over KMnO4, MeOH was distilled over magnesium and iodine, and DMF was distilled over CaH2. 1,1,1,3,3,3-hexafluoro-2-propanol (HFIP) was used as is from commercial sources.”
Reviewer’s comment: I am a little unclear how you sublimed an oil. "An analytical sample was obtained by sublimation at...) (line 386 and other locations).
Answer: The sentence was corrected:
Old text: “An analytical sample was obtained by sublimation at 65 °C (0.75 Torr).”; “An analytical sample was obtained by two-fold sublimation at 65 °C (0.75 Torr).”
Corrected version: “An analytical sample was obtained by vacuum distillation at 65 °C (0.75 Torr).”; “An analytical sample was obtained by two-fold vacuum distillation at 65 °C (0.75 Torr).”
Reviewer’s comment: Consider depositing your crystal structures to the CCDC and providing CCDC numbers for readers.
Answer: Crystal structures of 2p and 2t were deposed to the CCDC and corresponding CCDC numbers were depicted in Scheme 3. Corresponding CCDC numbers can be also found in SI (see S122).
Reviewer’s comment:
Comments on the Quality of English Language:
In the abstract make sure results are presented clearly. You shouldn't be talking about "the proposed electrochemical approach (line 17)" you should be stating your results. Also the phrase: "which guarantees outstanding reproducibility" (line 22) is awkward. You should just state a key result that demonstrates your newly developed procedure is scalable.
References need to be reformatted to be before commas, periods, etc.
Do not use the word etc. (line 34)
Do not use the phrase "by one of the authors of this paper" (line 73). Say something like Churakov et al.
Use previously reported instead of known previously (line 143). You don't have to state that "all others were synthesized for the first time" this should be implied.
Don't say Quite interesting results were obtained from the experiment with ad divided electrochemical cell" (line 214). You can't define results as interesting.
In the second part of our study synthesized.....were discovered as a new class of fungicides. (line 236-237) This is phrased awkwardly.
rotary evaporated (line 323) is not correct language. Solvent removed in vacuo.
Answer: We thank the Reviewer for careful reading of our manuscript. Additional language polishing was made.